# Using Biochar and Nanobiochar of Water Hyacinth and Black Tea Waste in Metals Removal from Aqueous Solutions

Fathy Elbehiry [1], Marwa Darweesh [2], Fathia S. Al-Anany [2], Asmaa M. Khalifa [3], Aliaa A. Almashad [3], Hassan El-Ramady [4], Antar El-Banna [5], Vishnu D. Rajput [6], Hanuman Singh Jatav [7] and Heba Elbasiouny [2,*]

1   Basic and Applied Science Department, Higher Institute for Agricultural Co-Operation, Shubra El-Kheima 13766, Egypt
2   Department of Environmental and Biological Sciences, Home Economics Faculty, Al-Azhar University, Tanta 31732, Egypt
3   Food Science and Technology Department, Home Economics Faculty, Al-Azhar University, Tanta 31732, Egypt
4   Soil and Water Department, Faculty of Agriculture, Kafrelsheikh University, Kafr El-Sheikh 33516, Egypt
5   Department of Genetics, Faculty of Agriculture, Kafrelsheikh University Kafr, El-Sheikh 33516, Egypt
6   Academy of Biology and Biotechnology, Southern Federal University, 344090 Rostov-on-Don, Russia
7   Soil Science and Agricultural Chemistry, Sri Karan Narendra Agriculture University, Jaipur 303329, India
*   Correspondence: hebaelbasiouny@azhar.edu.eg

**Abstract:** The treatment of heavy metal-contaminated water is challenging. The use of nanomaterials from many environmental wastes is promising for removing metals and contaminants from aqueous solutions. This study is novel in using nanobiochar of water hyacinth (WH) and black tea waste (TW) as a promising approach to water decontamination owing to its unique properties that play an effective role in metal adsorption. The mono- and multi-adsorption systems of cadmium (Cd), chromium (Cr), and nickel (Ni) on biochar and nanobiochar of water hyacinths (BWH and NBWH) and black tea waste (BTW and NBTW) were investigated in this study as potential low-cost and environmentally friendly absorbents for the removal of previously mentioned heavy metals (HMs) from aqueous solutions. The WH and TW were collected from the locality, prepared, and kept until used in the experiment. Nanobiochar was prepared by grinding, characterizing, and storing in airtight containers until used. A batch experiment was designed in mono- and competitive systems to study the adsorption equilibrium behavior of HMs on biochar and nanobiochars. The Freundlich and Langmuir isotherm models were fitted to the mono- and competitive-adsorption equilibrium results. The Freundlich isotherm model provided a better fit. Furthermore, it was noticed that NBWH and NBWT efficiently removed the Cd in the mono-system by ≥99.8, especially in the smaller concentration, while NBWT and BTW removed ≥99.8 and 99.7% in the competitive system, respectively. In the mono- and competitive systems, the nanobiochars of NBTW removed more than 98.8 of Cr. The sorbents were less efficient in Ni removal compared to Cd and Cr. However, their effectiveness was very high also. The results revealed that Cd was the highest metal removed by sorbents, nanobiochars were better than biochars to remove the HMs, and the results also indicated that co-occurrence of multi-metals might fully occupy the adsorption sites on biochars and nanobiochars.

**Keywords:** water contamination; water hyacinths; black tea waste; biochar; nanobiochar; aqueous solutions

## 1. Introduction

Water is a necessary and priceless resource. It is needed for the requirements of human life, such as food production or even direct ingestion [1]. Thus, water contamination is a serious environmental problem that impacts human health, water quality, and wildlife [2]. Treatment of generated wastewater in developing countries is a serious concern, particularly

for those who take important factors such as cost, sustainability, reusability, and recycling into account. Furthermore, the high costs associated with many of the available methods for reducing environmental pollution can be a burden to society, particularly on low-income households. Because of the high expense of the many types of equipment and chemicals utilized, wastewater treatment is becoming a significant problem [3,4]. Although many traditional methods are applied to treat water, they are neither cost-effective nor environmentally friendly. Thus, innovative green materials and technologies are currently being deployed. It is important to understand the ideas behind green chemistry and green technology. Alternative, more affordable, and environmentally responsible approaches to water management that enhance water quality and protect the environment should be considered in developing countries [1].

Heavy metals (HMs) pollution in particular is a worldwide environmental phenomenon, and it is one of the most severe and noticeable issues in water bodies. Moreover, it has been globally increasing at an alarming rate [5–7]. Heavy metals are generated in the environment mainly from various human activities, such as fertilizers and pesticide application, utilization of sewage in agricultural irrigation, mining, electronic manufacturing discharge, and recycling and dumping of electronic waste. These sources have increased the amount of metals in the water bodies or wastewater discharged into the aquatic environment, which leads to water contamination with metals [8–12]. Heavy metals such as Cd, Cr, and Ni in the aquatic environment are non-biodegradable and can be bioaccumulated and biomagnified in food chains. Even at low concentrations and because of their toxicity, they can seriously harm human health [6,9,13,14]. Thus, HMs remediation of contaminated water is necessary. Over the past few decades, many remediation methods have been conducted for contaminated water, such as C adsorption, ion exchange, precipitation, flocculation, coagulation, membrane filtration, technologies, biodegradation, etc. However, some of these procedures have limitations and disadvantages, such as high costs and low efficiency [14–20]. Therefore, it is essential to design more affordable wastewater treatment methods, taking into account the requirement for domestic wastewater pre-treatment and the expense of installing current technology. Additionally, extensive research has been conducted to identify low-cost, readily accessible materials from agricultural wastes as sorbents to reduce the issues associated with using previous traditional techniques [4]. Thus, there is a need for more eco-friendly and sustainable substitutes for conventional remediation methods [21].

Adsorption is considered one of the most promising techniques for HMs removal from aqueous solutions owing to its simplicity, flexibility, and high efficiency [2]. Bio-sorbents are environmentally safe, readily available in huge quantities, and low-cost products that are easy to operate and highly efficient. They are usually used in water treatments. Biosorbents are biological materials that can accumulate HMs from the environment through metabolically mediated or spontaneous physicochemical uptake pathways or as a property of specific kinds of inactive, non-living microbial biomass that bind and concentrate HMs from even a very diluted aqueous solution. Biochar is one of the most common bio-sorbents [15,22–24]. It is a C-rich adsorbent that has a high ability and efficacy to adsorb HMs from the environment owing to its high specific surface area, well-built porous construction, and high thermal stability. Biochar can be acquired by biological waste pyrolysis at temperature ranges from 350° to 800° under oxygen-limited conditions [5,9,25,26].

The application of water hyacinths (WH) and waste tea (WT) leaves to remove HMs from contaminated water through adsorption is an eco-friendly and cost-effective approach [5,16,25]. Water hyacinth is a highly invasive aquatic plant that has spread in many geographic regions. It has caused substantial damage to nearby water quality, among many other problems [25]. Biochar of WH has recently shown the potential to adsorb metals from contaminated water. The high pyrolytic process temperature alters the surface biochar properties, allowing for greater HM sorption [23,25]. In addition, TW contains all the properties of a good feedstock for biochar production. Therefore, its use for removing or immobilization HMs can be a suitable approach to contaminated water management

and environmental remediation [5]. However, the capacity of raw biochar is low for the elimination of high-concentration pollutants. Thus, biochar modification can not only alter the surface area, porosity, and exterior functional groups but also allocate bulk characteristics such as reactivity, charge density, surface functional groups biocompatibility, and stability [27]. Although there are environmentally friendly methods for treating wastewater, nanotechnology has been recognized to be one of the most promising methods that could play a significant role in solving many issues relating to water purification and quality [28,29]. Nanoparticles have been widely used in bioremediation because of their high specific surface area, which enhances the removal reactions [30,31]. Nanobiochar has recently gained popularity among engineered biochars due to its beneficial physio-chemical properties [7,32]. Nanobiochar has been considered an approach for environmental sustainability [33].

Indeed, WH and TW are not new species used for HMs removal. They exist in abundance in many countries, but they are negligible and cause many problems related to waste management. Additionally, the new forms of WH and TW, such as biochar and nanobiochar, have unique and distinct properties (such as large surface area) that have proven efficient in removing HMs from aqueous solutions. However, these new forms are not studied very well. However, most water decontaminated studies by biochar used other materials. In addition, there are insufficient studies about the use of nanobiochar in this objective. Thus, the novelty of this study represents the focus on the study of new forms of WH and TW that were not studied enough, especially the nanobiochar of these negligible wastes. Furthermore, it highlights and compares the effectiveness of biochar and nanobiochar of WH and TW to remove heavy metals from contaminated water. Hence, because of this reason, in addition to a need to find more ecofriendly materials, this study aims to study the adsorption affinity of biochar and nanobiochar of WH and TW towards Cd, Cr, and Ni in mono- and multi-metals' competitive adsorption conditions. Thus, this research will contribute to a better understanding of using biochar or nanobiochar of some environmental wastes in water decontamination experiments.

## 2. Materials and Methods

### 2.1. Biochar and Nanobiochar Preparation

Water hyacinths were obtained from Metyazed canal in Kafr Elsheikh governorate, Egypt. The whole plant was washed 5 to 7 times with tap water, followed by rinsing in deionized water. The cleaned plants were spread on aluminum trays and dried at 70 °C (BD, Binder, NY, USA) until the constant weight of dried plant samples and used to produce biochar. Black tea was collected as waste from households, washed with tap and deionized water, spread on aluminum trays, and dried at 70 °C (BD, Binder, NY, USA) until the constant weight of black tea samples. The samples were converted into biochar by pyrolysis under a limited oxygen condition using a furnace (ZBX1, China) at 500° for 4 h in a porcelain airtight container.

Nanobiochar samples were prepared by grinding the samples into nano-size by ceramic mortar for about 9 h manually and kept in airtight containers until characterization, analysis, and handling in the experiment.

### 2.2. Biochar and Nanobiochar Characterization

Biochar morphology was studied using a transmission electron microscope (TEM) (JEOL, JEM2100, Tokyo, Japan). The Fourier transform infrared (FTIR) spectra of specimens were observed by Jasco FT/IR 4100 spectrometer in the wavelength range of 4000–400 $cm^{-1}$, and all the samples were equipped on KBr tablets. Biochar was analyzed for some physiochemical properties (pH (1:10 biochar: water) by a pH meter (JENWAY 3510, UK, EC (1:5 biochar: water) and TDS (1:5 biochar: water) by an EC-meter (Mi170, Milwaukee, Italy). The organic matter (OM) content was assessed by the method of Walkley [34]. Available trace metals (Cd, Cr, and Ni) were extracted using ammonium bicarbonate–

diethylenetriaminepentaacetic (AB-DTP) [35] and measured by atomic absorption spectrometry (AAS) (GBC Avanta E, Victoria, Australia).

*2.3. Sorption Experiments*

2.3.1. Mono-Metal Sorption System

The batch mono-adsorption experiment was accomplished in 50 mL Erlenmeyer flasks containing 20 mL of the tested metal solutions at initial concentrations of (5, 10, 20, and 40 mg L$^{-1}$). Biochar (BWH and BTW by addition rate of 0.4 g) and nanobiochar (NBWH and NBTW by addition rate of 0.2 g) were mixed with 20 mL of metal solutions. Before adding the sorbents, the pH was adjusted to 5.0 with 1 M hydrochloric acid (HCl) and 1 M sodium hydroxide (NaOH). The sorption investigations were conducted on a thermostatic for 24 h at room temperature. Some drops of toluene were added to suppress microbial activity. After adsorption, the suspensions were centrifuged at 3000 rpm for 10 min. The supernatants were collected in separate clean test tubes and filtered through Whatman No. 42 filter paper. The supernatant of Cd, Cr, and Ni concentrations was measured by atomic absorption spectrometry (serial No. A5616; GBC Avanta E, Victoria, Australia).

2.3.2. Competitive Sorption System

A batch equilibrium experiment was also conducted using Cd, Cr, and Ni in a competitive sorption system with metal concentrations of 5, 10, and 20 mg L$^{-1}$ for all metal solutions as follows: biochar (addition rate of 0.4 g) and nanobiochar (addition rate of 0.2 g) were mixed with 20 mL of metal solutions, respectively, as nitrate salts in 50 mL centrifuge tubes for 24 h in a reciprocating shaker at room temperature. After equilibration, the samples were centrifuged and filtered, and the metal concentrations were determined as previously described.

The influence of adsorbents on the behavior of metal ions adsorption was investigated. The *qe* equation, Equation (1), was applied for calculating the metal adsorption capacity in the adsorbent unit (*qe*, mg/g) [14,36]:

$$qe = (C0 - Ce)\ V/m \tag{1}$$

where C0 and Ce separately are the initial and equilibrium concentrations of Cd, Cr, and Ni (mg L$^{-1}$). V is the solution volume (L), whereas *m* is the biochar mass load (g). The metal removal from the aqueous solution was assessed, as in Equation (2):

$$\text{Removal percentage \%} = ((C0 - Ce)/C0) * 100 \tag{2}$$

where C0 and Ce are the initial and equilibrium concentration, respectively (mg L$^{-1}$).

The partition coefficient (PC, L g$^{-1}$) was calculated as the following equation (Equation (3)) [14]:

$$PC = q/Cf \tag{3}$$

where q (mg g$^{-1}$) is the adsorption capacity, while Cf (mg L$^{-1}$) is its corresponding final HM concentration, respectively.

The adsorption results were fit into Freundlich and Langmuir models as in Equations (4) and (5) [14,36].

$$\text{Freunlich model } \log C_s = \log K_f + \left(\frac{1}{n}\right) \log C_e \tag{4}$$

$$\text{Langmuir model } \frac{Ce}{qe} = \frac{1}{(KL\ qm)} + \frac{Ce}{qm} \tag{5}$$

where Cs is the equilibrium adsorption capacity, K$_f$ is the constant of Freundlich, the value ($\frac{1}{n}$) is the adsorption linearity, Ce is the equilibrium metal concentration in the aqueous phase (mg L$^{-1}$), qe is the equilibrium metal concentration adsorbed on biochar (mg g$^{-1}$),

$K_L$ is the constant referring to the bonding energy of adsorption (L mg$^{-1}$), and qm is the maximum capacity of adsorption (mg g$^{-1}$).

### 2.3.3. Quality Control and Statistical Analysis

All the equipment used was calibrated and uncertainties were estimated. Internal and external quality controls were operated in the laboratory based on ISO/IEC 17025- 2017 requirements for laboratory accreditation. In all estimations, blanks, triplicate samples of each metal in the extract, and the analysis of certified reference material for each metal obtained from Phenova-certified reference material (WS0718; September 2018) were regularly involved for quality control. The standard of recovery metals ranged from 95 to 106%. The average relative standard between replications (RSD) was <5% in most samples and above 5% in a few instances. These values were not included in the statistical analyses. The detection limits (LoD) obtained for Cd, Cr, and Ni were 2.7, 3.4, and 1.4 $\mu$L$^{-1}$, respectively. The results of HMs and pH were within the accepted limits of the proficiency testing provider. The data collected were analyzed statistically with SPSS 22 software (IMB SPSS Statistics Software, Version 20; IBM, Armonk, NY, USA). Statistical analysis was performed with one-way analysis of variance (ANOVA). A Duncan's test of multiple ranges was run for mean on comparison of the treatments, data variability was recorded as the standard deviation, and $p \leq 0.05$ was considered to be statistically significant.

## 3. Results and Discussion

### 3.1. Biochar and Nanobiochar Characterization

#### 3.1.1. Physiochemical Properties and Metal Concentrations of Biochar and Nanobiochar

Table 1 presents the characteristics of sorbents, indicating that all sorbents have an alkaline pH, and the highest pH value was recorded with BTW followed by NBTW. The EC, TDS, and OM values were higher in BWH than in other sorbents even NBWH. Biochar's metal composition has been stated to be a valuable predictor for determining its nature [14]. The values of the studied cations were higher in BWH and NBWH compared to in BTW and NBTW. Table 1 also shows the results of metal composition analysis for biochar and nanobiochar samples. All the investigated metals (i.e., Cd, Cr, and Ni) were not detected under their detection limits in these samples, indicating that these sorbents will not represent any source for these metals in the aqueous solution.

**Table 1.** Physiochemical properties and metal concentrations of biochar and nanobiochar.

| | pH | EC dS m$^{-1}$ | TDS (g L$^{-1}$) | OM (g kg$^{-1}$) | Total Cations (mg kg$^{-1}$) | | | | Available Heavy Metals (mg kg$^{-1}$) | | |
|---|---|---|---|---|---|---|---|---|---|---|---|
| | | | | | Na | K | Ca | Mg | Cd | Cr | Ni |
| BWH | 9.90 | 8.10 | 4.00 | 682 | 57.3 | 9.0 | 19.8 | 13.4 | <0.01 | <0.02 | <0.03 |
| BTW | 10.71 | 3.31 | 1.6 | 430 | 23.5 | 7.2 | 10.4 | 8.8 | <0.01 | <0.02 | <0.03 |
| NBWH | 9.95 | 7.8 | 3.7 | 611 | 48.8 | 10.2 | 16.3 | 12.8 | <0.01 | <0.02 | <0.03 |
| NBTW | 10.2 | 2.8 | 1.4 | 402 | 22.4 | 6.6 | 9.4 | 7.2 | <0.01 | <0.02 | <0.03 |

BWH, biochar of water hyacinth; BT W, biochar of tea waste; NBWH, nanobiochar of water hyacinth; NBTW, nanobiochar of tea waste; EC, electrical conductivity; TDS, total dissolved salts; OM, organic matter; nd, non-detected.

#### 3.1.2. Biochar and Nanobiochar Characterization by TEM

As indicated in Figure 1, the TEM image of biochar and nanobiochar of BTW, NBTW, BWH, and NBWH are given. High-resolution electron microscopy may be used to identify morphology, size, and composition [37,38]. TEM images show the connected pores on the surface of biochars and nanobiochars. The NBTW ranged in size between 26.3 and 38.4 nm, while the NBWH ranged between 28.2 and 45.8 nm.

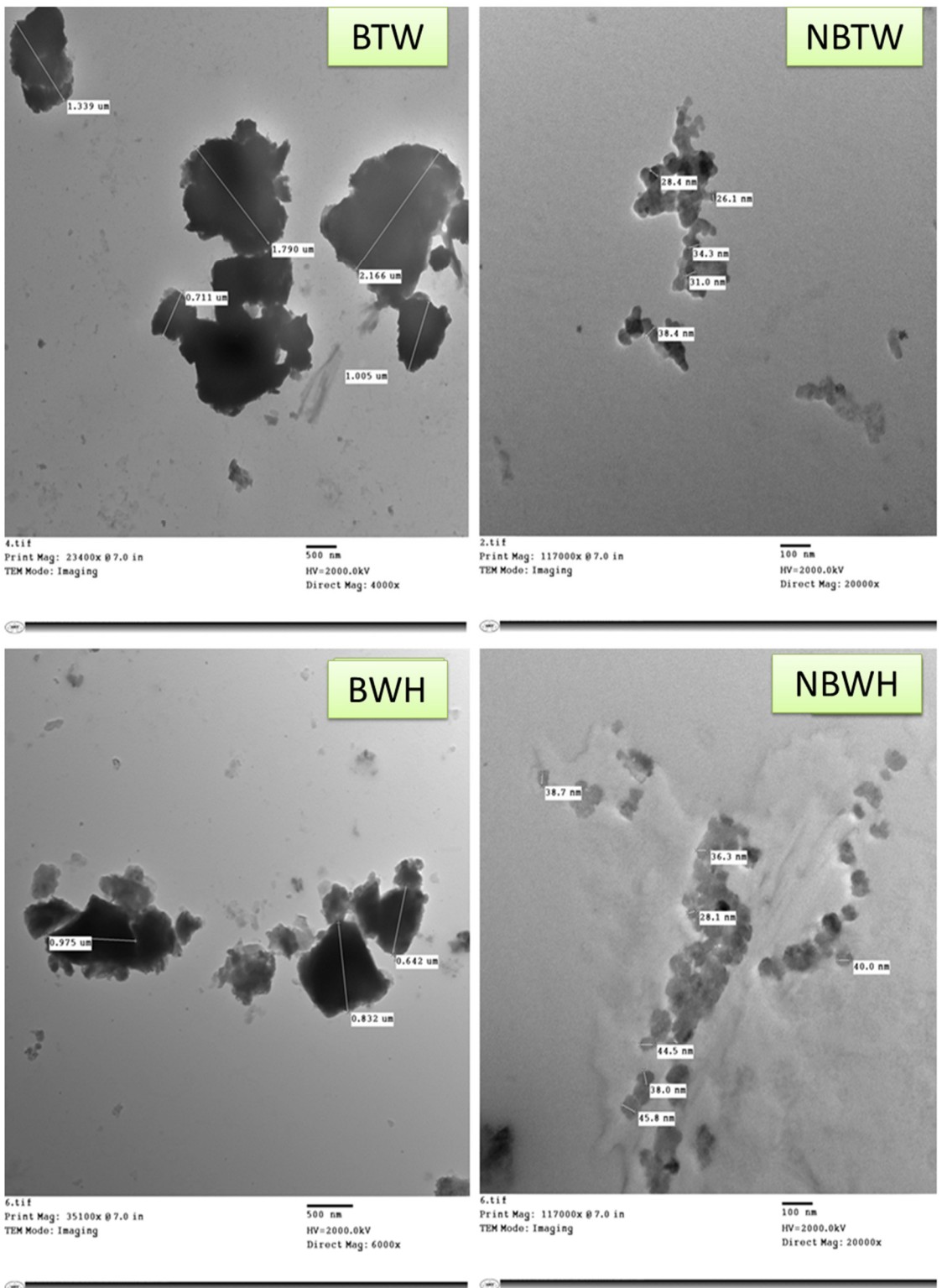

**Figure 1.** TEM analysis of biochar and nanobiochar of BTW, NBTW, BWH, and NBWH.

### 3.1.3. Biochar Functional Groups Characterization by FTIR

Before adsorption, the FTIR spectra of BWH revealed a weak CH band at 1100 cm$^{-1}$, whereas it appeared at 1500 cm-1 and 3300 cm$^{-1}$ at BTW (Figure 2), indicating a single bond belonging to alkanes. A medium stretching OH band appeared at 3500 cm$^{-1}$ and 3900 cm$^{-1}$ for BWH and BTW, respectively, which corresponds to alcohol compounds. Specific adsorption of BTW involves the surface complexation amongst functional groups

and HMs in wastewater, and hence, strong and constant compounds are created; C, H, O, N, and S are the key elements in BTW. This is why the functional groups on the BTW surface area are primarily made of surface hydroxyl groups, which convey surrounding ions together and provide protons to the solution. When electron-pair acceptor ions are difficult, surface complexes are created between the metal ions and hydroxyl groups [39].

The aromatic CC band was also evident at 880 cm$^{-1}$ for WH and 1000 cm-1 for BTW, indicating that alkanes were present. The presence of the functional group C-O of hemicellulose, cellulose, and lignin was observed in the C-O band at 1850 cm$^{-1}$ in both WH and BTW, indicating the substantial presence of the functional group C-O of hemicellulose, cellulose, and lignin. Hemicellulose and lignin serve as matrix and encrusting components, respectively, and create a skeleton around cellulose. The lignocellulosic nature makes a particular chemical makeup, with a limited number of inorganic components and a relatively high proportion of volatile organic chemicals. This produce compounds with a well-defined pores structure/network and a large surface area as a result of their chemical composition, which plays an important role in adsorption [40]. The FTIR study also revealed that the COO band, which belongs to carbon dioxide, was visible at 2350 cm$^{-1}$ for BWH only, whereas the NO band, which belongs to nitro compounds, was visible at 1500 cm$^{-1}$ for BTW only. Weng et al. [41] demonstrated the presence of OH and C-H groups in the WT cellulosic material by FTIR. The presence of the lignin aromatic bond (C-C) could explain the -C=C and -C=O groups. FTIR was employed by Munene et al. [42] to corroborate the presence of O-H, C=O, and C-H in BWH. FTIR of NBWH and NBTW showed a little difference from BWH and WT concerning the functional group.

After mono-Cd adsorption, the FTIR spectra of BWH reveal weak C-Cl, C=C, C-O, OH, and S-S bands as well as the removal of C-H and O-C-O bands (Figure 3A). In addition, modest S-S, N-O, C=C, and OH bands occur in the FTIR spectra of BWH after mono-Ni adsorption (Figure 3B) as well as the elimination of C-H and O-C-O, C-O bands. With the C-O and O-C-O bands lacking, the FTIR spectra of BWH following mono-Cr adsorption show a faint S-S, alkayne C-H, C=C, methyne C-H stretch, and OH stretch (Figure 3C). The existence of faint S-S, C-l, C-F, carbonate ion, methyne C-H, and OH bands can be seen in the FTIR spectra of BWH after adsorption of competitive metals (Figure 3D), while the disappearance of N-O is revealed.

After mono-Cd adsorption, the FTIR spectra of BTW reveal stretching O-H, C-H, N=C=O, S-O, and S-S and S-S bands as well as the removal of S-H and C-H bands (Figure 4A). In addition, modest S-O, C=C, and OH bands occur in the FTIR spectra of WH after mono-Ni adsorption (Figure 4B) as well as the elimination of S-H band. The FTIR spectra of WH following mono-Cr adsorption show a faint S-S, alkayne C-H, C=C, methyne C-H stretch, and OH stretch (Figure 4C). The existence of faint S-S, methyne C-H, and OH bands can be seen in the FTIR spectra of BTW after adsorption of competitive metals (Figure 4D), while the disappearance of N-O is revealed.

The FTIR spectra of nano BTW following adsorption of competitive metals (Figure 5A) reveals weak bond of C-H, N-O (which do not appear in Figure 1; water hyacinth), C-O, C-H, and OH bands (Figure 4A) as well as a weak C-H, C=C, and S=O (which do not appear also in Figure 1; black tea).

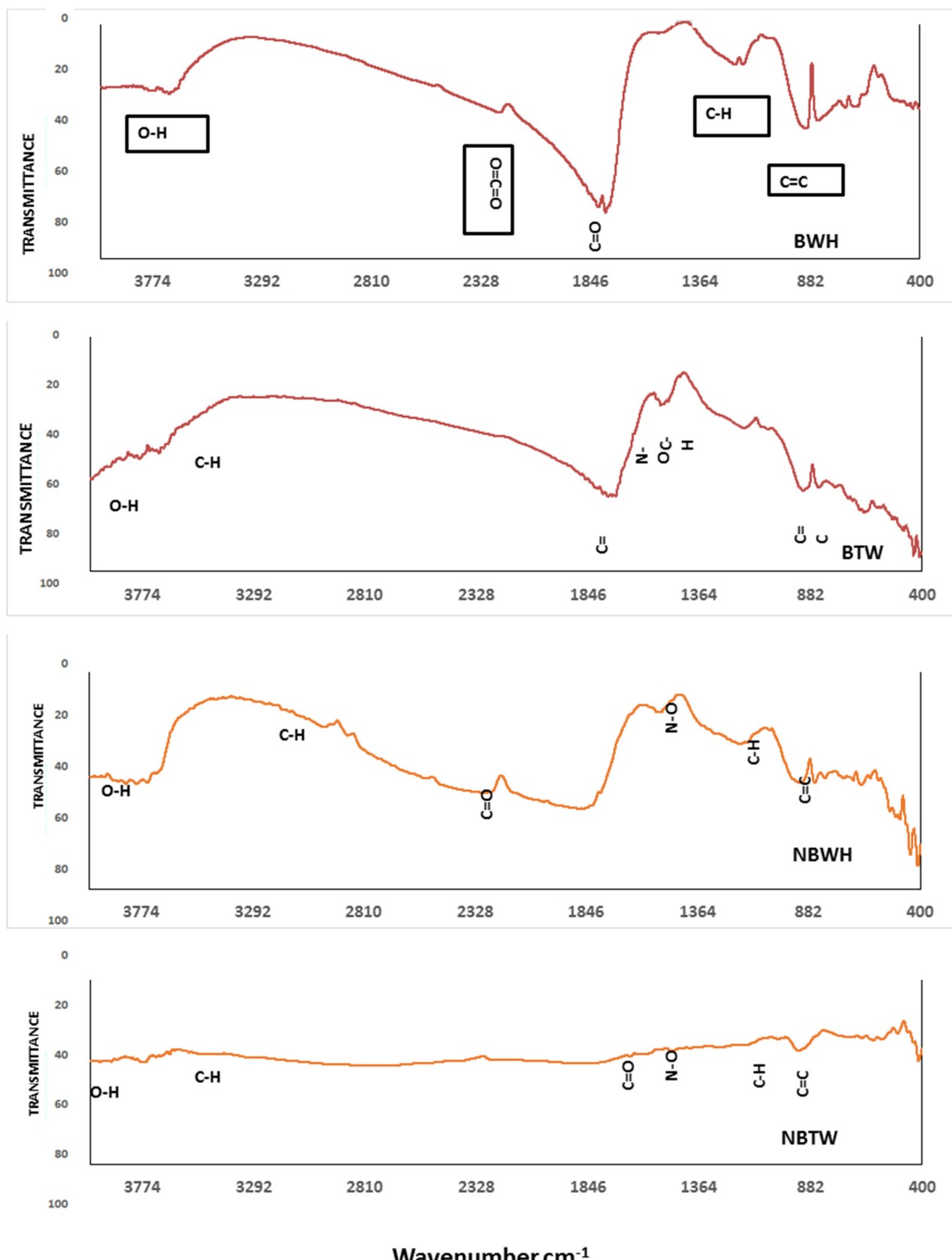

**Figure 2.** FTIR spectra of water hyacinth and black tea biochar and nanobiochar before adsorption.

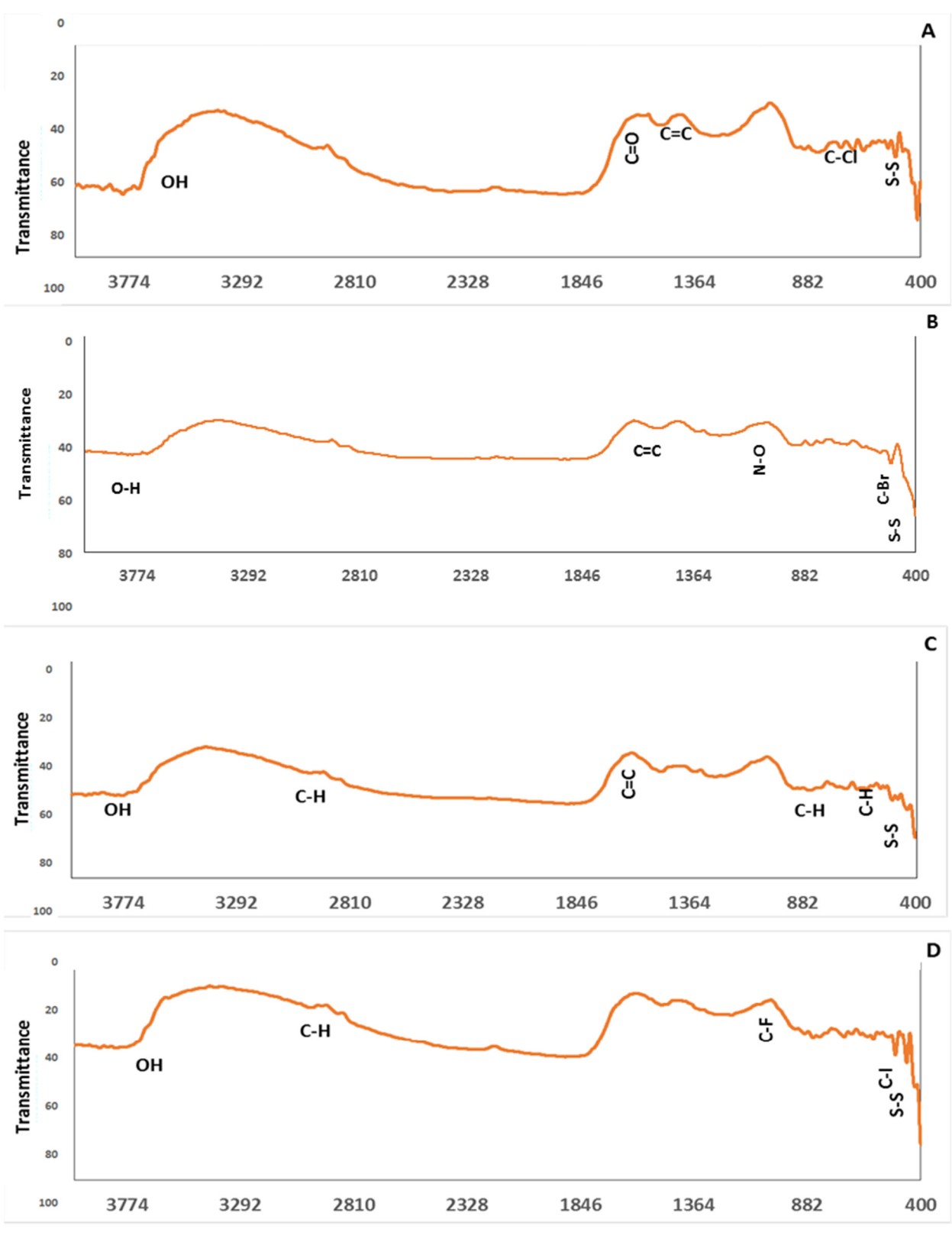

**Figure 3.** FTIR spectra of water hyacinth biochar after adsorption. (**A**) BWH 20 mg L$^{-1}$ mono-Cd, (**B**) BWH 20 mg L$^{-1}$ mono-Ni, (**C**) BWH 20 mg L$^{-1}$ mono- Cr, and (**D**) BWH 20 mg L$^{-1}$ competitive metals.

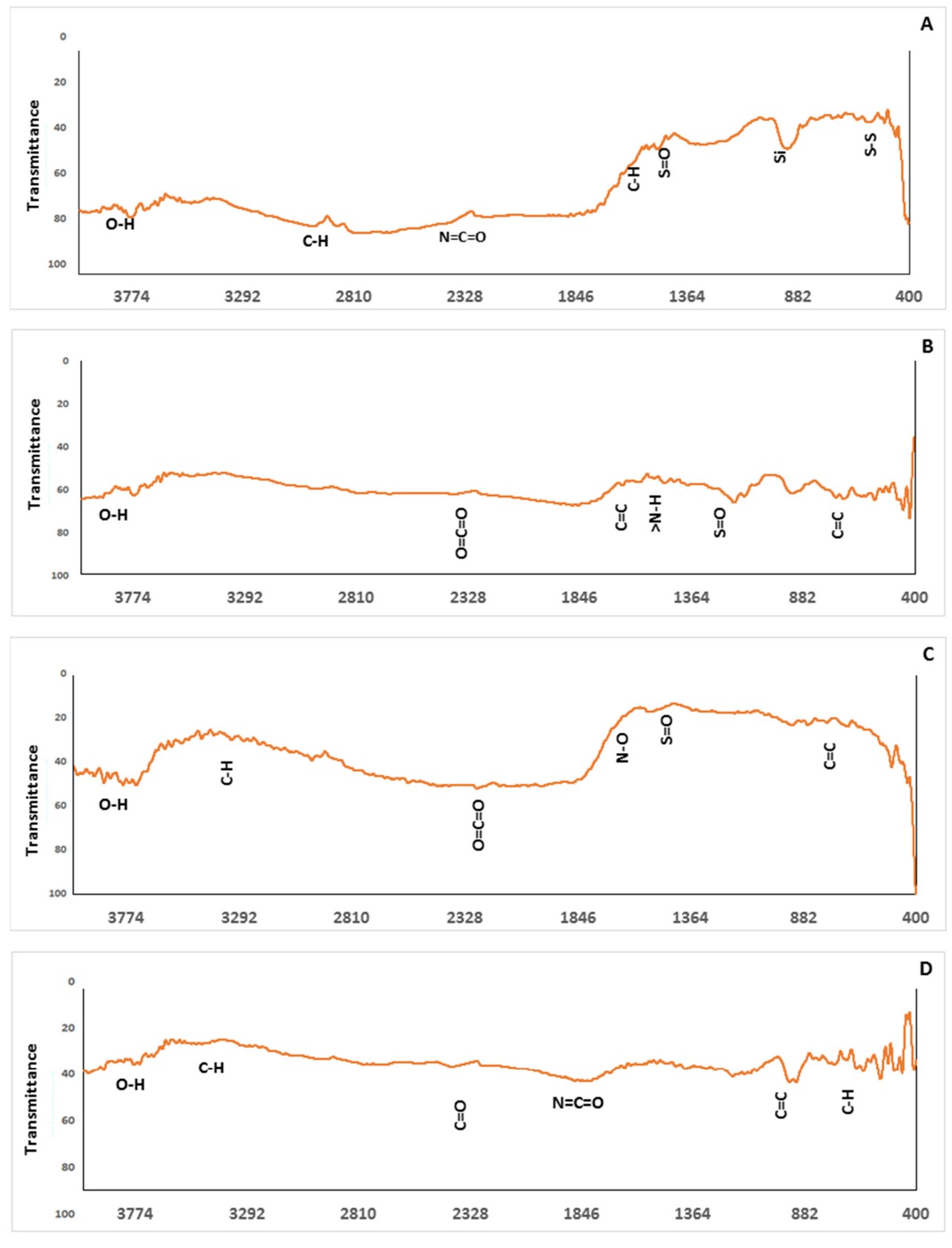

**Figure 4.** FTIR spectra of waste tea biochar after adsorption. (**A**) BTW 20 mg L$^{-1}$ mono-Cd, (**B**) BTW 20 mg L$^{-1}$ mono-Ni, (**C**) BTW 20 mg L$^{-1}$ mono- Cr, and (**D**) BTW 20 mg l$^{-1}$ competitive metals.

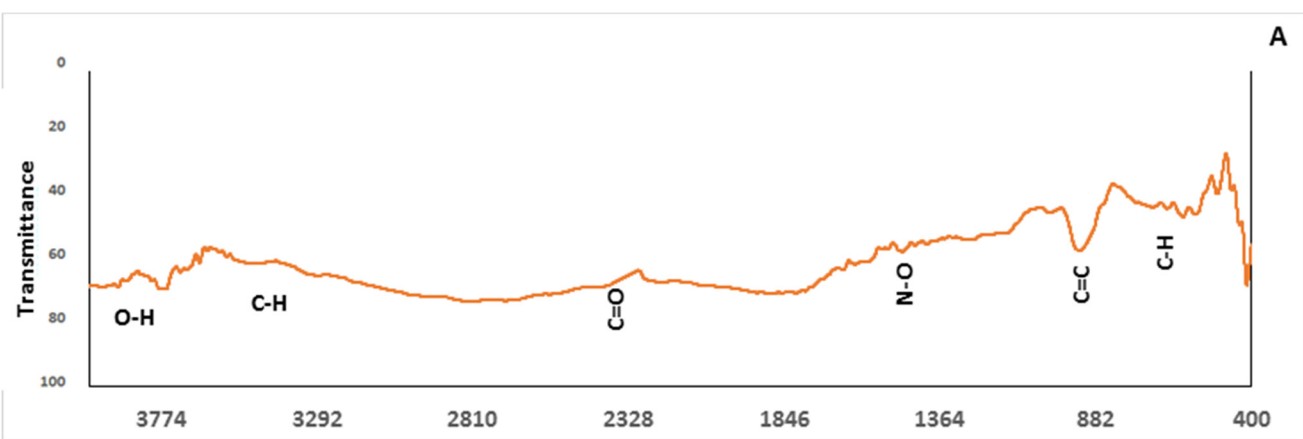

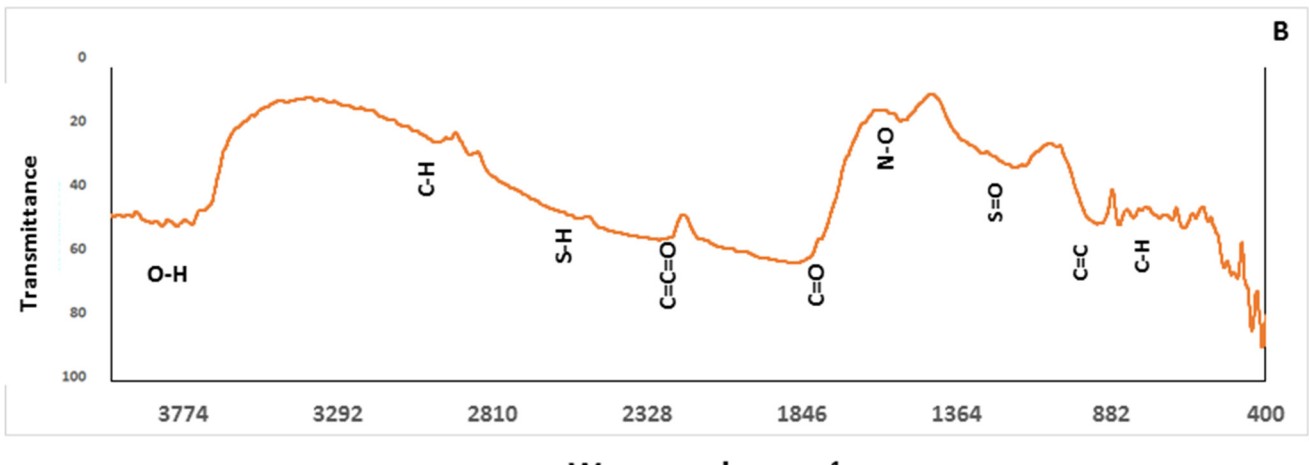

**Figure 5.** FTIR spectra of nano-water hyacinth and nano-waste tea biochar after adsorption. (**A**) BTW 20 mg L$^{-1}$ competitive metals; (**B**) BTW 20 mg L$^{-1}$ competitive metals.

### 3.2. Adsorption Efficiency and Isotherm Models' Characteristics

#### 3.2.1. Removal Efficiency

Table 2 shows the removal efficiency (RE) of mono and competitive adsorption concentrations of Cd, Cr, and Ni at equilibrium by tested sorbent materials. In mono and competitive systems, both forms (biochar and nanobiochar) of WH and TW demonstrated very high HMs removal efficiencies of more than 85%. Nanobiochars showed higher potential metal removal than biochars. The highest adsorption capacity for Cd, under mono-adsorption conditions, was recorded in NBWH and NBTW ($\geq$99.8% removal efficiency), especially with initial Cd concentrations of 5 to 20 mg L$^{-1}$, indicating that nanobiochar forms had a high affinity for Cd. However, NBTW (99.8% RE) showed the highest adsorption capacity of Cd under the competitive adsorption conditions at all concentrations. The efficiency in Cd removals by studied biochar generally may be attributed to their high pH, which is supported also by Herath et al. [43], who observed that the higher pH of the solution, the high Cd adsorbed on the biochar. Furthermore, they found that using lake water at 8.8 pH resulted in higher Cd removal, which they attributed to precipitation of Cd(OH)$_2$ above pH 6. Ouyang et al. [44] conducted that the hydroxyl groups on the surface of sorbent played a key role in the HMs removal from the solution. Moreover, the Cd removal efficiency by BWH is attributed to the presence of some functional groups, as indicated by [42,45]. The highest adsorption capacity of Cr, under mono and competitive adsorption conditions, was recorded in NBWH (98.8% and 99.2% RE, respectively), both at Cr concentration of 40 mg L$^{-1}$. Herath et al. [43] used commercial biochar and KOH-activated biochar to adsorb Cr, Pb, and Cd from aqueous solutions. They stated that Cr(VI) adsorption to biochar

can be attributed to one of the following mechanisms: (a) direct electrostatic attractions between (+)-charged surface sites of biochar and (–)-charged Cr oxyanions and (b) Cr(VI) reduction into Cr(III) by oxidation of surface groups of biochar at C via water-mediated transfer of H. However, because their pH was 2 or higher, Cr did not adsorb significantly on biochars. According to Herath et al. [43], Cr(VI) is more mobile and soluble in water than Cr(III), and it is a known carcinogen that harms the human respiratory and urinary systems. Khalil et al. [46] used WT biochar for Cr (VI) removal from wastewater with an effective RE of 99.3% and indicated that -OH and COO- were involved in the sorption according to FTIR analysis.

**Table 2.** Mono and competitive (multi-metals) removal percentage (RE % ) and the partition coefficient (PC, L $g^{-1}$) of Cd, Cr, and Ni for the tested sorbent materials.

| Adsorbent Martial | Initial Metal Conc. mg $L^{-1}$ | Cd | | | | Cr | | | | Ni | | | |
|---|---|---|---|---|---|---|---|---|---|---|---|---|---|
| | | Mono | | Competitive | | Mono | | Competitive | | Mono | | Competitive | |
| | | RE (%) | PC | RE (%) | PC | RE (%) | PC | RE (%) | PC | RE (%) | PC | RE (%) | PC |
| BWH | 5 | 97.4 | 0.95 | 99.5 | 4.98 | 98.4 | 1.54 | 98.2 | 1.36 | 98.8 | 2.02 | 96.2 | 0.63 |
| | 10 | 95.8 | 0.56 | 99.2 | 2.92 | 97.8 | 1.09 | 97.5 | 0.96 | 93.1 | 0.33 | 94.6 | 0.44 |
| | 20 | 96.3 | 0.65 | 99.4 | 4.14 | 96.8 | 0.75 | 97.1 | 0.82 | 94.6 | 0.43 | 94.8 | 0.45 |
| | 40 | 97.2 | 0.86 | 99.4 | 4.48 | 97.0 | 0.81 | 97.3 | 0.89 | 95.0 | 0.47 | 95.2 | 0.49 |
| BTW | 5 | 88.4 | 0.19 | 99.7 | 9.23 | 95.5 | 0.53 | 97.0 | 0.81 | 95.6 | 0.45 | 87.9 | 0.18 |
| | 10 | 85.8 | 0.15 | 99.3 | 3.31 | 88.4 | 0.19 | 96.9 | 0.78 | 93.0 | 0.27 | 85.9 | 0.15 |
| | 20 | 85.3 | 0.14 | 99.4 | 3.82 | 91.3 | 0.26 | 96.4 | 0.67 | 94.7 | 0.46 | 87.1 | 0.17 |
| | 40 | 85.9 | 0.15 | 99.5 | 4.93 | 91.7 | 0.28 | 96.6 | 0.71 | 93.8 | 0.39 | 87.7 | 0.18 |
| NBWH | 5 | 99.8 | 26.27 | 99.7 | 18.47 | 98.6 | 3.52 | 99.2 | 6.20 | 98.3 | 2.94 | 98.2 | 2.76 |
| | 10 | 99.6 | 12.77 | 99.3 | 6.62 | 97.6 | 2.03 | 98.5 | 3.28 | 98.0 | 2.39 | 97.3 | 1.80 |
| | 20 | 99.6 | 14.03 | 99.4 | 7.64 | 98.5 | 3.23 | 98.6 | 3.52 | 98.3 | 2.94 | 98.3 | 2.81 |
| | 40 | 99.7 | 17.23 | 99.5 | 9.39 | 98.8 | 3.97 | 98.9 | 4.33 | 98.6 | 3.61 | 98.7 | 3.78 |
| NBTW | 5 | 99.8 | 24.95 | 99.8 | 24.95 | 98.5 | 3.28 | 98.5 | 3.35 | 97.0 | 1.62 | 95.9 | 1.17 |
| | 10 | 99.7 | 15.33 | 99.8 | 26.98 | 98.2 | 2.71 | 98.2 | 2.65 | 95.4 | 1.04 | 94.9 | 0.92 |
| | 20 | 99.5 | 10.65 | 99.8 | 20.36 | 98.5 | 3.34 | 98.6 | 3.59 | 97.0 | 1.63 | 87.0 | 0.33 |
| | 40 | 99.6 | 11.90 | 99.8 | 22.75 | 98.7 | 3.74 | 98.8 | 4.01 | 97.3 | 1.83 | 88.4 | 0.38 |

BWH, biochar water hyacinth; BTW, biochar tea waste; NBWH, nanobiochar water hyacinth; NBTW, nanobiochar tea waste.

The highest adsorption capacity of Ni, under mono and competitive adsorption conditions, was recorded in BWH and NBWH ($\geq$98.80, and 98.70% RE of mono and competitive Ni, respectively) but with different concentrations. This may be attributed to the presence of -COO- functional group, and it is a key in $Ni^{2+}$ biosorption [7,47]. The previous results indicated that WH in both forms was better than BTW in metal sorption as well; NBWH had the highest affinity to Cd, Cr, and Ni in an aqueous solution. Munene et al. [42] studied in a batch experiment the absorbance of $Cd^{2+}$, $Cr^{3+}$, $Pb^{2+}$, $Ni^{2+}$, and $Zn^{2+}$ on BWH powder. They noticed a reduction, shift, or disappearance of the spectra based on the metal. All metal ions were observed to show reduced adsorption bands on the BWH, which confirms that functional groups are involved in adsorption. Peng et al. [45] studied the adsorption of As, Cd, Cu, and Pb on BWH and confirmed that the presence of bonds such as C-O, C-C, O-H, and C-H showed to help remove HMs from aqueous solution and accumulating in WH tissues. In addition, in our study, the RE was higher under competitive adsorption conditions of Cd and Cr. On the contrary, Abdin et al. [14] found that metals' RE was higher under mono-adsorption conditions although they studied different types of biochar. The RE in competitive adsorption was in the order of Cd (99.8%) for NBTW > Ni (98.7%) for NBWH and >Cr (99.2%) for NBWH.

### 3.2.2. Partition Coefficient (PC)

It is stated in [14,48] that the comparison of performance among different adsorbents is critical to evaluating their real performance metrics. Thus, in this study, the PC of each sorbent was calculated as a function of the initial metal concentration (Table 2). The obtained PC values demonstrated that the performance of the investigated sorbents was

higher in the competitive metal adsorption except in NBWH in Cd and Ni than that of the mono-metal adsorption, following the same illustrated trend in RE%. The PC values showed a tendency to decline at the initial metal concentration. The lower PC values at a lower initial metals concentration could be ascribed to less favorable sorption sites, while the higher PC values at a lower initial metals concentration are primarily because of the more available sorption sites with higher selectivity and binding energies at low initial metals concentration. Correlatively, the RE behavior and trends correspond to PC [14]. Hence, PC in conjunction with RE as a function of initial metal concentration can be utilized to better understand the efficiency of different sorbents to remove metal from an aqueous solution. Based on the PC values, NBTW had better performance in mono-Cd (26.98 at 5 mg L$^{-1}$) > NBWH for Cr-competitive (6.20 at 5 mg L$^{-1}$)> NBHW for Ni-competitive (3.78 at 40 mg L$^{-1}$).

### 3.2.3. Freundlich and Langmuir Isotherm Models

The Freundlich isotherm (Equation (4)) occurs on heterogonous surfaces and results by assuming that adsorption sites are spread exponentially about adsorption heat. Additionally, it implies that the stronger active sites are first occupied and that the binding intensity reduces as site occupancy increases. It is useful for non-ideal adsorptions on dissimilar surfaces as well as multilayer adsorption [36,49,50]. The parameters matched to Freundlich and Langmuir isotherm equations are presented in Table 3. Fierro et al. [51] indicated that "n" is an empirical parameter that represents the energetic heterogeneity of the sites of adsorption. Adsorption is typically satisfactory if the Freundlich constant "n" values range from 1–10, which is almost achieved in our results in Cd only with all sorbents except BWH. The KF is the coefficient of the Freundlich model, which is related to the adsorption degree. The higher its value, the larger the affinity of adsorbate–adsorbent. It is used very well for the comparison of the adsorption values of different metals on biochars [52]. The KF values of Cd were higher in NBTW for Cd-competitive followed by NBWH for Cd-mono compared to that of Cr and Ni, which proves a stronger binding affinity of Cd to the studied sorbents than that of Cr and Ni. It is also observed that KF values were almost higher in competitive systems than in mono systems in Cd and Cr. This implies that the co-existing of the multi-metals could fully occupy the adsorption sites on biochars and nanobiochars. The R$^2$ values of the Freundlich equation, which were close to one, demonstrate that ions were bound to the heterogonous surface of investigated biochars and nanobiochars by multilayer adsorption and favorably sorbed on the high-active sites. Additionally, it indicates the adsorption isotherm is very well-modelled by Freundlich [51,53]. This varied between the studied metals and sorbents.

**Table 3.** Isotherm models' parameters for Cd, Cr, and Ni adsorption on biochars and nanobiochars.

| Treat-ments | Initial Conc. mg L$^{-1}$ | Cd | | | | | | Cr | | | | | | Ni | | | | | |
|---|---|---|---|---|---|---|---|---|---|---|---|---|---|---|---|---|---|---|---|
| | | Langmuir | | | Freundlich | | | Langmuir | | | Freundlich | | | Langmuir | | | Freundlich | | |
| | | Q Max | KL | R$^2$ | 1/n | K$_F$ | R$^2$ | Q Max | KL | R$^2$ | 1/n | K$_F$ | R$^2$ | Q Max | KL | R$^2$ | 1/n | K$_F$ | R$^2$ |
| BWH | Mono | 1.28 | 0.80 | 0.96 | 0.92 | 0.69 | 0.93 | 0.98 | 1.75 | 0.98 | 0.74 | 0.76 | 0.99 | 0.47 | 5.67 | 0.85 | 0.53 | 0.46 | 0.83 |
| | Comp. | 1.34 | 3.98 | 0.94 | 0.94 | 3.51 | 0.94 | 1.14 | 1.31 | 0.98 | 0.82 | 0.82 | 0.99 | 1.33 | 0.51 | 0.97 | 0.89 | 0.48 | 0.98 |
| BTW | Mono | 1.58 | 0.13 | 0.99 | 0.90 | 0.17 | 0.99 | 0.61 | 1.05 | 0.89 | 0.73 | 0.30 | 0.89 | 1.40 | 0.33 | 0.94 | 0.98 | 0.38 | 0.94 |
| | Comp. | 0.66 | 16.95 | 0.91 | 0.71 | 2.29 | 0.89 | 3.72 | 0.22 | 0.99 | 0.92 | 0.70 | 0.99 | 3.37 | 0.05 | 0.99 | 1.00 | 0.17 | 0.99 |
| NBWH | Mono | 1.65 | 18.19 | 0.93 | 0.80 | 8.87 | 0.93 | 3.18 | 1.17 | 0.93 | 1.02 | 3.17 | 0.89 | 19.62 | 0.15 | 0.98 | 1.10 | 3.41 | 0.97 |
| | Comp. | 1.31 | 17.03 | 0.91 | 0.71 | 4.48 | 0.90 | 1.79 | 3.91 | 0.94 | 0.82 | 3.06 | 0.94 | 6.62 | 0.42 | 0.94 | 1.12 | 3.18 | 0.89 |
| NBTW | Mono | 1.71 | 0.02 | 0.97 | 0.71 | 6.19 | 0.98 | 18.33 | 0.18 | 0.98 | 1.08 | 3.65 | 0.98 | 4.48 | 0.37 | 0.94 | 1.05 | 1.56 | 0.92 |
| | Comp. | 11.18 | 2.32 | 0.99 | 0.92 | 17.60 | 0.99 | 23.51 | 0.14 | 0.98 | 1.11 | 3.97 | 0.97 | 1.54 | 0.89 | 0.99 | 0.58 | 0.62 | 0.96 |

BWH, biochar water hyacinth; BTW, biochar tea waste; NBWH, nanobiochar of water hyacinth; NBTW, nanobiochar of tea waste; Comp., competitive.

The Langmuir isotherm describes the formation of adsorbed monolayers of solute and is based on the hypotheses that metal ions are adsorbed on a fixed number of well-defined surface sites, with each site hosting one ion, all sites being powerfully equal, and the ions not interacting. This isotherm means a decreasing adsorption capacity; once monolayer coverage is obtained, the solute transport no longer has a major effect on sorption. When the surface is filled by a monolayer of the adsorbate, maximal adsorption occurs [36]. The obtained data supported the previous statement that most of the studied systems did not fit the Langmuir model because the $R^2$ values of the Langmuir model that fit were lower than their corresponding Freundlich model fitting in some conditions, which assumes the multilayer adsorption on sorbent surfaces in these conditions [14].

The Cd adsorption isotherm (mono or competitive) (Figure 6) can be explained as the H-type in all studied adsorbents, as described by Giles et al. [54], which suggests that the sorption sites on the surface of adsorbents were not fully occupied by Cd, and there is difficulty in forming a second layer due to charge repulsion. The shapes of isotherm can be described as L-type in Cr-mono with BWH and BTW and H-type with NBWH and NBTW, while in Cr-competitive, it can be described by H-types in all conditions. Isotherm shapes can be described by H-type in Ni with all nano forms either mono or competitive. It can be labelled as L-type in Ni-mono and Ni-competitive with WH, while it can be characterized by L-type in Ni-competitive with BWH and C-type in Ni-competitive with BTW only. Giles et al. [54] reported that the examples of C-type adsorption inorganic and biological substrates that illustrate the continuing permeation of micropores in the substrate and increasing expansion of the substrate network with the amount of adsorbents. L-type is the more widespread type and can occur with adsorption of either monodisperse or aggregated substrates. It occurs when the adsorption sites are few and commonly separated, and the surfaces have large hydrophobic areas. This means that the increasing rate of adsorbed amount for a given metal tended to decline with the equilibrium concentrations [14]. Furthermore, L-type reflects no surface precipitation [55]. Generally, the results indicate that co-occurrence of multi-metals might fully occupy the adsorption sites on biochars and nanobiochars. However, in competitive systems, the HMs sorption does not only depend on the surface characteristics of the sorbents but also on the properties of metal ions [14]. In the presence of many ions of metal, the competition among them occurs on the coordination sites that are presented on the adsorbent surface [56]. Xiao et al. [57] attributed high-adsorption performances to the larger atomic weight of the metal, with its ionic radius associated with smaller hydrated radius, Misono softness, and first hydrolysis constant, which all causes Cd to have a better affinity to investigated sorbents than other metals via complexation reactions on the surfaces of biochar and nanobiochar. Ion exchange, precipitation or co-precipitation, electrostatic attraction surface complexation with the functional groups, in addition to physical adsorption may participate in HMs absorption on biochar [14].

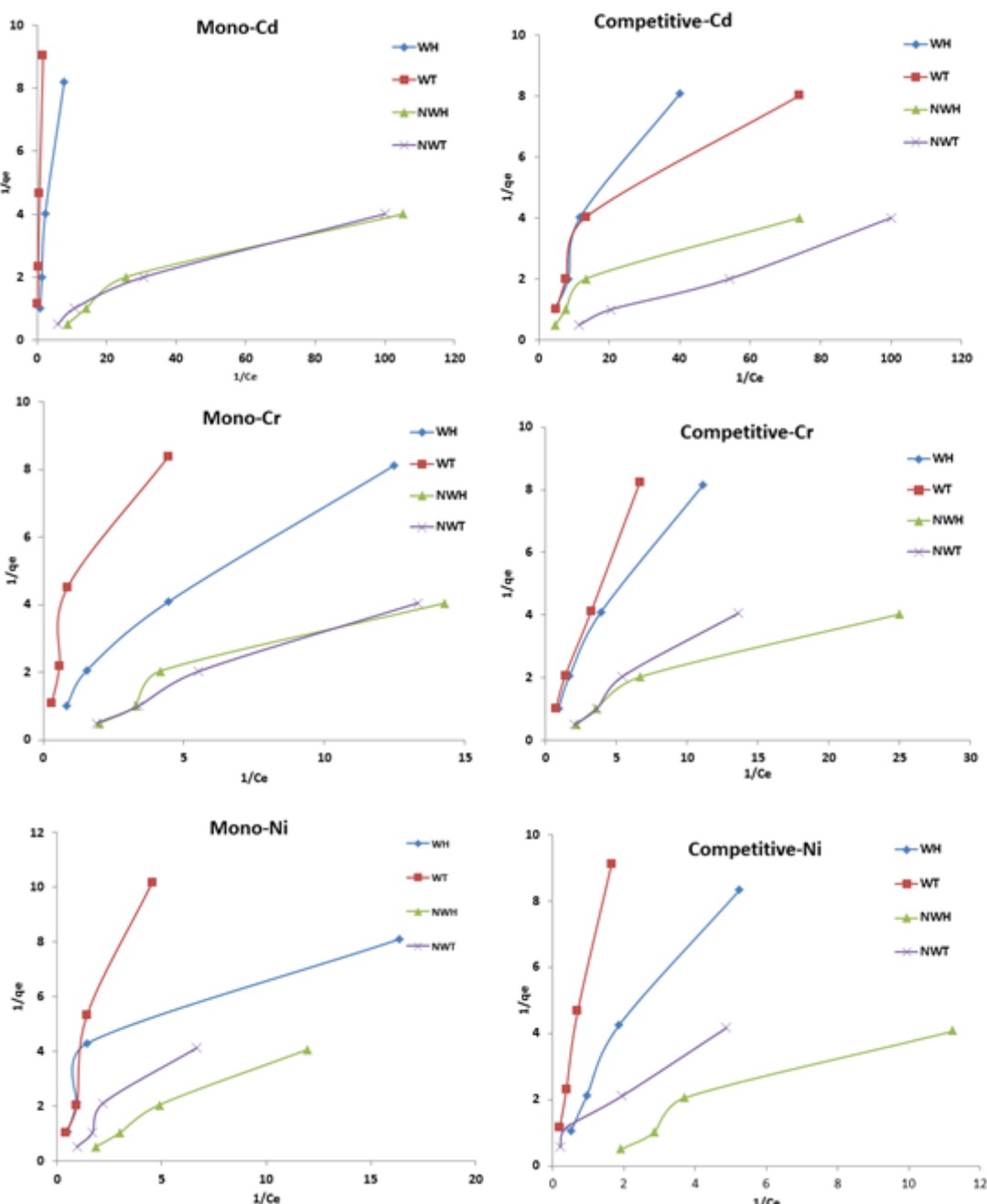

**Figure 6.** Mono- and competitive adsorption isotherms of Cd, Cr, and Ni for biochars and nanobiochars Table 2.

## 4. Conclusions

Biochar and nanobiochar of water hyacinths and black tea waste were used as eco-friendly materials to remove Cd, Cr, and Ni in mono- and competitive systems of the aqueous solution. It is suitable for use especially in developing countries because of its low-cost preparation and its benefits as sustainable and eco-friendly materials. Thus, this approach is promising and novel, especially using nanobiochar in water purification and decontamination. The sorbents removed the studied metals efficiently from aqueous solution, especially Cd from contaminated solution. In addition, the nanobiochars were more highly efficient in this purpose than biochar. The removal efficiency of the sorbents was higher than 85% in all studied sorbents either by biochar or nanobiochar. The Ni was less-removed by the studied sorbents. The metals tended to be more adsorbed in the

multi-systems and mono-systems in some conditions under the studied sorbents. The data that were best-fitted to the Freunlich isotherm adsorption model indicated that the metals were adsorbed in multilayers by different mechanisms. However, more studies are needed to better understand the utilization of nanobiochar in the removal of toxic metals from contaminated water and the effect of various factors on the adsorption of these metals. Furthermore, the potential utilizing of these materials on a large scale in wastewater treatment plants and in making filters for the factories to reduce the pollutants reaching the water bodies should be considered in future research, especially for studying the flow rate and time.

**Author Contributions:** Conceptualization, F.E., M.D. and H.E.; methodology, F.E., M.D., and H.E.; software, F.E. and H.E.; validation, F.E., H.E.-R., V.D.R., and H.E.; formal analysis, F.E., M.D., F.S.A.-A., A.M.K., A.A.A., A.E.-B. and H.E.; investigation, F.E., M.D.; F.S.A.-A. A.M.K., A.A.A., A.E.-B. and H.E.; resources, F.E., M.D., F.S.A.-A., A.M.K., A.A.A., A.E.-B., H.E.-R., V.D.R., H.S.J. and H.E.; writing—original draft preparation, F.E., M.D., and H.E.; writing—review and editing, F.E., M.D., F.S.A.-A., A.M.K., A.A.A., A.E.-B., H.E.-R., V.D.R., H.S.J. and H.E.; supervision, F.E., H.E.-R., V.D.R. and H.E. All authors have read and agreed to the published version of the manuscript.

**Funding:** This research received no external funding.

**Institutional Review Board Statement:** Not applicable.

**Informed Consent Statement:** Not applicable.

**Data Availability Statement:** Once accepted, data will be made available by the authors.

**Acknowledgments:** The authors acknowledge support by the the laboratory «Soil Health» of the Southern Federal University with the financial support of the Ministry of Science and Higher Education of the Russian Federation, agreement No. 075-15-2022-1122.

**Conflicts of Interest:** The authors declare no conflict of interest.

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
