# Peer review of "Using Biochar and Nanobiochar of Water Hyacinth and Black Tea Waste in Metals Removal from Aqueous Solutions"

_sustainability, doi:10.3390/su141610118_

Round 1

Reviewer 1 Report

Review Report for the review paper entitled

 “Water Hyacinth and Back Tea Waste Biochar and Nanobiochar 2 for Metals Removal from Aqueous Solutions”. 

 (Manuscript Number: Susutainability-1798233) 

Major comments                                                                           

1.        In Introduction section, Authors wrote that Many remediation approaches for contaminated water have been conducted, such as C adsorption, ion exchange, precipitation, flocculation, coagulation, membrane filtration, technologies, biodegradation

Correct this sentence and the reference for these claims is missing. I suggest authors to include the following references and cite accordingly,

a)      Summary on Adsorption and Photocatalysis for Pollutant Remediation: Mini Review. Journal of Encapsulation and Adsorption Sciences, 2018, 8, 225-255. DOI.org/10.4236/jeas.2018.84012

b)      Synthesis and Characterization of Ti-Fe oxide nanomaterials for lead removal. Journal of Nanomaterials, (Volume 2018, Article ID 9651039, 10 pages). Hindawi Publication.

DOI.org/10.1155/2018/9651039  

2.        In equations 2 and 3

Authors have used two different notations for equilibrium concentrations- Ce and Ceq. Better to use single notation.

3.        In Results and discussion section- Author wrote that “High-resolution electron microscopy may be used to identify morphology, size, composition, and crystallinity”

Authors did not present SAED pattern of samples to confirm the crystallinity. In addition, authors did not provide XRD pattern for the synthesised samples.

4.        Figure 2.

Authors have not used proper scale on x-axis. Why did authors select very limited range?

5.    Overall English and grammar must be improved.

Minor comments/ grammatical/typographic errors

1.      Page 1, line 22, 36, 35- nanobichar- modify as nanobiochar

2.      Page 1, line 30- Sorbets- modify as sorbents.

3.      Page 2, line 96- 70 degrees Celsius– use common unit as 70 oC.

4.      Page 4, Table 1 - Total catoins (mg kg-1)– correct this.

5.       Page 5- - The NWT was 26.3–38.4 nm in size, while the 193 NWH was 28.2–45.8 nm in size.  Rephrase this sentence.

6.      Page 6 - the FTIR spectra of WH revealed a weak CH band at 1100 cm-1, 198 whereas it appeared at 1500 cm-1 and 3300 cm-1 at WT (Fig. 2),- unit must be represented as cm-1

7.     

8.      ….

9.     

10.  …many more…

Author Response

Dear Reviewer, 

Thanks so much for your valuable and constructive comments that will improve the paper. The response for your comments are as follows: 

Major comments                                                                           

  1. In Introduction section, Authors wrote that Many remediation approaches for contaminated water have been conducted, such as C adsorption, ion exchange, precipitation, flocculation, coagulation, membrane filtration, technologies, biodegradation

Correct this sentence and the reference for these claims is missing. I suggest authors to include the following references and cite accordingly,

  1. a)      Summary on Adsorption and Photocatalysis for Pollutant Remediation: Mini Review. Journal of Encapsulation and Adsorption Sciences, 2018, 8, 225-255. DOI.org/10.4236/jeas.2018.84012
  2. b)      Synthesis and Characterization of Ti-Fe oxide nanomaterials for lead removal. Journal of Nanomaterials, (Volume 2018, Article ID 9651039, 10 pages). Hindawi Publication.

DOI.org/10.1155/2018/9651039  

Response:

Thanks for the comment, the sentence has been corrected. The references are already existed (14-18), however both required references are inserted  (Line 76).

  1. In equations 2 and 3

Authors have used two different notations for equilibrium concentrations- Ce and Ceq. Better to use single notation.

Response:

The notation Ceq is corrected to Ce  (Line 90 and 191)

  1. In Results and discussion section- Author wrote that “High-resolution electron microscopy may be used to identify morphology, size, composition, and crystallinity”

Authors did not present SAED pattern of samples to confirm the crystallinity. In addition, authors did not provide XRD pattern for the synthesised samples.

 Response:

The word crystallinity (Line 244) is deleted and actually, this work was self-funded and there wasn’t external funding, as well these analyses here are very costly so it is so difficult to do them. However, we will consider it in the future works 

        Figure 2.

Authors have not used proper scale on x-axis. Why did authors select very limited range? Change made in the figures

  1. Overall English and grammar must be improved. Change made through the manuscript

Minor comments/ grammatical/typographic errors

  1. Page 1, line 22, 36, 35- nanobichar- modify as nanobiochar. Response: The only mention of this word was in line 22 and it is corrected
  2. Page 1, line 30-Sorbets- modify as sorbents. Response: The word is corrected in this line and through the manuscript
  3. Page 2, line 96-70 degrees Celsius– use common unit as 70 oC. Response: It is corrected in lines 93, 141 and 144
  4. Page 4, Table 1- Total catoins (mg kg-1)– correct this. Response: It is corrected in Table 1
  5. Page 5- - The NWT was 26.3–38.4 nm in size, while the 193 NWH was 28.2–45.8 nm in size.  Rephrase this sentence. Response: The sentence is rephrased in line 244-246
  6. Page 6 - the FTIR spectra of WH revealed a weak CH band at 1100 cm-1, 198 whereas it appeared at 1500 cm-1 and 3300 cm-1 at WT (Fig. 2),- unit must be represented as vcm-1. Response: The unit is corrected it was a typing mistake as well as it is corrected in whole paragraph and through the manuscript.
  7. ….
  8. …many more…

Response: the paper was revised again to consider any of these issues, as well some comments of other reviewers are made and considered.

My best regards, 

Heba Elbasiouny

Reviewer 2 Report

The current manuscript in title: (Water Hyacinth and Back Tea Waste Biochar and Nanobiochar for Metals Removal from Aqueous Solutions) illustrates the possibility of biochar and nano-biochar of water hyacinths (WH and NWH) and black tea waste (WT and NWT) as potential low-cost and eco-friendly absorbents for the removal of heavy metals (HMs) from aqueous solutions. However, several points are important to be addressed before going to possible publication in this high-quality journal. Also, the authors need to address all points in the revision stage for broad range readers understanding. Moreover, additional data are necessary to provide.

The title should be improved.

English grammar and style must be revised; many sentences have no meaning or sense. The manuscript needs thorough revision to improve the text quality and readability of the work.

The introduction need to improve with some recent refs., especially in the application of low-cost and eco-friendly absorbents. Authors may use the following recommended refs. ( https://doi.org/10.3390/gels8050310 ; https://doi.org/10.3390/ma15113922  ; https://doi.org/10.3390/polym14071375 )

Lines 99-101: “Nanobiochar samples… “. More  characterizations still needed such as  Dynamic Light Scattering (DLS), BET.

 In all the manuscript, please, change "degrees Celsius) and “ ºC “ to “°C”.

Please, improve the quality of Figs.

Author Response

Dear reviewer, 

Thanks so much for you valuable and constructive comments that will improve the manuscript. The response to the comments are as follows:

Reviewer 2

Comments and Suggestions for Authors

The current manuscript in title: (Water Hyacinth and Back Tea Waste Biochar and Nanobiochar for Metals Removal from Aqueous Solutions) illustrates the possibility of biochar and nano-biochar of water hyacinths (WH and NWH) and black tea waste (WT and NWT) as potential low-cost and eco-friendly absorbents for the removal of heavy metals (HMs) from aqueous solutions. However, several points are important to be addressed before going to possible publication in this high-quality journal. Also, the authors need to address all points in the revision stage for broad range readers understanding. Moreover, additional data are necessary to provide.

The title should be improved. Response: Change made

English grammar and style must be revised; many sentences have no meaning or sense. The manuscript needs thorough revision to improve the text quality and readability of the work. Response: Change made through the manuscript

The introduction need to improve with some recent refs., especially in the application of low-cost and eco-friendly absorbents. Authors may use the following recommended refs. ( https://doi.org/10.3390/gels8050310 ; https://doi.org/10.3390/ma15113922  ; https://doi.org/10.3390/polym14071375 ) Response: Change made in the introduction

Lines 99-101: “Nanobiochar samples… “. More  characterizations still needed such as  Dynamic Light Scattering (DLS), BET. Response:

Actually, this work was self-funded and there wasn’t an external fund this analysis here is very costly so we couldn’t do them. However, we will consider it in the future works

 In all the manuscript, please, change "degrees Celsius) and “ ºC “ to “°C”.

 Response: It is corrected in all the manuscripts (lines 93, 141, and 144)

Please, improve the quality of Figs. Change made in the figures.

My best regards, 

Heba Elbasiouny

Reviewer 3 Report

Dear authors,

Many thanks for submitting the manuscript entitled ‘Water Hyacinth and Back Tea Waste Biochar and Nanobiochar for Metals Removal from Aqueous Solutions’ to be considered for publication in the journal Sustainability. This is an important topic for research, especially with the ever-crescent industrialization of LMIC’s followed by organic and chemical water pollution issues. The manuscript is well written and very well structured, it has strong statistical analysis and strong technical aspects and details of the analysis. I think the manuscript deserves to be published in the journal. However, I felt that there is a need to write more about where in the water/wastewater treatment stages this technology should/could be applied. Why this research is important? In which locations/settings/countries this should be applied? What is the availability of these raw materials worldwide? Need to mention the use of similar materials in the water treatment stages and how the new proposed materials' performance would compare to them? So, adding more background information about the usefulness of these materials in the Introduction, discussion and conclusion would improve the manuscript. Also, it would be great to explain why the use of this material would lead to a more sustainable world.  There are some comments and corrections detailed below which should be addressed before publication.   

Line number

Comment

18

Add the name of the metals, followed by their symbols

19

These acronyms are a bit confusing, shouldn’t they be  BWH (biochar water hyacinth); NBWH (Nanobiochar water hyacinth); BTW (Biochar tea waste) and NBTW (Nanobiochar tea waste)?  

22

Spelling error ; Biochar

23

Grinding, characterizing and storing…

30

NWT removed more …(delete ‘and’)

30

Which sorbets? This was not previously mentioned!

45

Electronic manufacturing effluents discharge? You should also mention the recycling and dumping of electronic waste  

46

Some of these sources are ‘diffuse  pollution and do not go into the wastewater streams, they go directly to the water sources

47

Add the names of the metals before their symbols, when thy first appear in the text  

50

They can also harm the environment, other aquatic life’s

52

Carbon absorption

54

High charge? You mean high costs?

59

Add a short definition of biosorbets’

92

Metyazed canal : Is this the name of the canal? If yes , delete ‘a’

94

Dried how? Give details about the drying method, including oven model name

94

What you mean ‘until constant weight’ is unclear ! Which weight?

97-98

Give technical details about the furnace

99-100

9 hours grinding! Constantly and interrupted?  Is this a person or machine working, give more details here

101

From a queues solution? What do you mean here?

cm−1: -1 should be superscript here!? This error occurs in other parts of the manuscript, correct all

107-108

This is a bit confusing: (pH (1:10 soil: water) by a pH-meter (JENWAY 3510, UK, EC (1:5 soil: water), and TDS (1:5 soil: water)

Need to explain better these phrase and  solution ratios, what you mean by soil here ? Was this the grinded biochar? If yes, need to change the names

109

By a method described by Walkley (1947) [21]

118

‘were equilibrated’ ? what you mean here? Mixed ? homogenized?

119

(HCl) ….sodium hydroxide (NaOH)

130

Equilibrated ?!

122

Add details about the centrifuge

131

Ad details of the reciprocating shaker

133

As described above (in 2.3.1)

173

P ≤ 0.05 , less or equal to 0.05 was considered significant

178

‘higher WH followed by NWT’ this is confusing! Something is missing here ! WT is the highest pH

178-179

This phrase here is also strange :’The EC, TDS, and OM values were higher  in WH and NWH than in the nano form of the same sorbet.’  Isn’t NWH the nano form of WH?

204

Spelling: This

277

Add author names here

283

Between positive… and negative….

315

What is the meaning of the PC acronym? Make sure that all acronyms are followed by their meaning when thy first appear in the manuscript

385

occur

398

were

406

Need to add more information about where and how these materials would be used in the water treatment stages

Author Response

Dear reviewer, 
Thanks so much for your valuable and constructive comments that will improve the manuscript. The response to the comments are as follows:

Reviewer 3

Dear authors,

Many thanks for submitting the manuscript entitled ‘Water Hyacinth and Back Tea Waste Biochar and Nanobiochar for Metals Removal from Aqueous Solutions’ to be considered for publication in the journal Sustainability. This is an important topic for research, especially with the ever-crescent industrialization of LMIC’s followed by organic and chemical water pollution issues. The manuscript is well written and very well structured, it has strong statistical analysis and strong technical aspects and details of the analysis. I think the manuscript deserves to be published in the journal. However, I felt that there is a need to write more about where in the water/wastewater treatment stages this technology should/could be applied. Why this research is important? In which locations/settings/countries this should be applied? What is the availability of these raw materials worldwide? Need to mention the use of similar materials in the water treatment stages and how the new proposed materials' performance would compare to them? So, adding more background information about the usefulness of these materials in the Introduction, discussion and conclusion would improve the manuscript. Also, it would be great to explain why the use of this material would lead to a more sustainable world.  There are some comments and corrections detailed below which should be addressed before publication.   

Thanks so much for your valuable comments. Some sentences are added in the introduction we hope to be a suitable answer to these questions. As well, the novelty and the importance of this research are added at the end of the introduction.

Line number

Comment

18

Add the name of the metals, followed by their symbols : change made : change made

19

These acronyms are a bit confusing, shouldn’t they be  BWH (biochar water hyacinth); NBWH (Nanobiochar water hyacinth); BTW (Biochar tea waste) and NBTW (Nanobiochar tea waste)?  : change made

22

Spelling error ; Biochar : change made

23

Grinding, characterizing and storing… change made

30

NWT removed more …(delete ‘and’) : change made

30

Which sorbets? This was not previously mentioned!  The word was corrected

45

Electronic manufacturing effluents discharge? You should also mention the recycling and dumping of electronic waste  change made

46

Some of these sources are ‘diffuse  pollution and do not go into the wastewater streams, they go directly to the water sources change made

47

Add the names of the metals before their symbols, when thy first appear in the text  change made in the first mention in the abstract

50

They can also harm the environment, other aquatic life’s this is mentioned in the text

52

Carbon absorption : C adsorption This mentioned in the reference

54

High charge? You mean high costs? change made

59

Add a short definition of biosorbets’ change made

92

Metyazed canal : Is this the name of the canal? If yes , delete ‘a’ change made in line 136

94

Dried how? Give details about the drying method, including oven model name change made

94

What you mean ‘until constant weight’ is unclear ! Which weight? change made

97-98

Give technical details about the furnace actually it is not in my lab and it is difficult to get this information now

99-100

9 hours grinding! Constantly and interrupted?  Is this a person or machine working, give more details here change made

101

From a queues solution? What do you mean here? change made and the words are deleted for more clearness

cm−1: -1 should be superscript here!? This error occurs in other parts of the manuscript, correct all change made

107-108

This is a bit confusing: (pH (1:10 soil: water) by a pH-meter (JENWAY 3510, UK, EC (1:5 soil: water), and TDS (1:5 soil: water)

Need to explain better these phrase and  solution ratios, what you mean by soil here ? Was this the grinded biochar? If yes, need to change the names change made

109

By a method described by Walkley (1947) [21] change made

118

‘were equilibrated’ ? what you mean here? Mixed ? homogenized? change made

119

(HCl) ….sodium hydroxide (NaOH) change made

130

Equilibrated ?! change made

122

Add details about the centrifuge actually it is not in my lab and it is difficult to get this information now

131

Ad details of the reciprocating shaker actually it is not in my lab and it is difficult to get this information now

133

As described above (in 2.3.1) change made

173

P ≤ 0.05 , less or equal to 0.05 was considered significant change made

178

‘higher WH followed by NWT’ this is confusing! Something is missing here ! WT is the highest pH change made

178-179

This phrase here is also strange :’The EC, TDS, and OM values were higher  in WH and NWH than in the nano form of the same sorbet.’  Isn’t NWH the nano form of WH? ’ change made

204

Spelling: This this word in not in 204

277

Add author names here There is no ref. in 277

283

Between positive… and negative…. Sorry this not understanding for us what you mean here

315

What is the meaning of the PC acronym? Make sure that all acronyms are followed by their meaning when thy first appear in the manuscript change made in the first mention in the title of this subsection

385

Occur this is come after it so it have to be occurs

398

Were it is not clear where to change

406

Need to add more information about where and how these materials would be used in the water treatment stages. Some information are added, however mo studies are needed

My best regards,

Heba Elbasiouny

Reviewer 4 Report

The present manuscript investigates the usage of mono- and multi-adsorption systems of Cd, Cr, and Ni on biochar and nano-biochar of water hyacinths (WH and NWH) and black tea waste (WT and NWT) as potential low-cost and environmentally friendly absorbents for the removal of heavy metals (HMs) from aqueous solutions. The results are interesting. On the other hand, the conclusion section needs to be rewritten Therefore, the manuscript should be reconsidered for publication in sustainability journal after major revision. The following comments should be taken into account.

1. The originality of the paper needs to be stated clearly. It is of importance to have sufficient results to justify the novelty of a high-quality journal paper. The Introduction should make a compelling case for why the study is useful along with a clear statement of its novelty or originality by providing relevant information and providing answers to basic questions such as: What is already known in the open literature? What is missing (i.e., research gaps)? What needs to be done, why and how? Clear statements of the novelty of the work should also appear briefly in the Abstract and Conclusions sections.

2. Authors should add more explanation about this investigation contribution in the introduction section.  

3. The introduction section should be updated with research articles that use different methods for the removal of heavy metals. The following articles may be used:

a)      Micromixing efficiency of particles in heavy metal removal processes under various inlet conditions, Water (Switzerland), 2019, 11(6), 1135.

b)      A computational tool for the estimation of the optimum gradient magnetic field for the magnetic driving of the spherical particles in the process of cleaning water, Desalination and Water Treatment, 2017, 99, 27–33.

c)      Numerical study of magnetic particles mixing in waste water under an external magnetic field, Journal of Water Supply: Research and Technology - AQUA, 2020, 69(3), 266–275.

d)      Mixing of Fe3O4 nanoparticles under electromagnetic and shear conditions for wastewater treatment applications, Journal of Water Supply: Research and Technology-Aqua, 2022, 71 (6), 671–681.

4. The experimental procedure should be analytically described.

5.  The conclusion section should be rewritten in order to include in detail the findings of the present work.

6. Lines 403 - 406 However, more studies are need to better understanding the utilizing of nanobiochar in removal of toxic metals from contaminated water and the effect of various factors on the adsorption of these metals.

Please include the parameters that affect the adsorption of the heavy metals by the nanobiochar. Why these parameters are not studied in this manuscript?

Author Response

Dear reviewer, 
Thanks so much for your valuable and constructive comments that will improve the manuscript. The response to the comments are as follows:

Reviewer 4

Comments and Suggestions for Authors

The present manuscript investigates the usage of mono- and multi-adsorption systems of Cd, Cr, and Ni on biochar and nano-biochar of water hyacinths (WH and NWH) and black tea waste (WT and NWT) as potential low-cost and environmentally friendly absorbents for the removal of heavy metals (HMs) from aqueous solutions. The results are interesting. On the other hand, the conclusion section needs to be rewritten Therefore, the manuscript should be reconsidered for publication in sustainability journal after major revision. The following comments should be taken into account.

Thanks so much for your valuable comments.

  1. The originality of the paper needs to be stated clearly. It is of importance to have sufficient results to justify the novelty of a high-quality journal paper. The Introduction should make a compelling case for why the study is useful along with a clear statement of its novelty or originality by providing relevant information and providing answers to basic questions such as: What is already known in the open literature? What is missing (i.e., research gaps)? What needs to be done, why and how? Clear statements of the novelty of the work should also appear briefly in the Abstract and Conclusions sections.

Thanks so much for your valuable comment, change made in the introduction

  1. Authors should add more explanation about this investigation contribution in the introduction section.Change made at the end of the introduction
  2. The introduction section should be updated with research articles that use different methods for the removal of heavy metals. The following articles may be used: change made in line 91 and some other recent references are added as recommended by other reviewers
  3. a)      Micromixing efficiency of particles in heavy metal removal processes under various inlet conditions, Water (Switzerland), 2019, 11(6), 1135.
  4. b)      A computational tool for the estimation of the optimum gradient magnetic field for the magnetic driving of the spherical particles in the process of cleaning water, Desalination and Water Treatment, 2017, 99, 27–33.
  5. c)      Numerical study of magnetic particles mixing in waste water under an external magnetic field, Journal of Water Supply: Research and Technology - AQUA, 2020, 69(3), 266–275.
  6. d)      Mixing of Fe3O4nanoparticles under electromagnetic and shear conditions for wastewater treatment applications, Journal of Water Supply: Research and Technology-Aqua,2022, 71 (6), 671–681.
  7. The experimental procedure should be analytically described.

It is already mentioned in section 2.3 and it is not clear to us what should we describe, so could you please explain more what should we add for more improvement.

  1. The conclusion section should be rewritten in order to include in detail the findings of the present work.

Change made

  1. Lines 403 - 406‘However, more studies are need to better understanding the utilizing of nanobiochar in removal of toxic metals from contaminated water and the effect of various factors on the adsorption of these metals.’

Please include the parameters that affect the adsorption of the heavy metals by the nanobiochar. Why these parameters are not studied in this manuscript? This is not included because this work was self-funded and there wasn't an external fund, so in the future, we will consider these parameters such as temperature, pH, time ….etc.

My best regards,

Heba Elbasiouny

Round 2

Reviewer 2 Report

The manuscript is accepted in its current form

Reviewer 4 Report

The manuscript can now be accepted for publication.

This manuscript is a resubmission of an earlier submission. The following is a list of the peer review reports and author responses from that submission.

Round 1

Reviewer 1 Report

This manuscript reports adsorption of heavy metals such as Cd, Cr and Ni by biochar and nano-carbon of Water Hyacinths and Black Tea waste. It shows relatively superior adsorption performance. However, there is still a lack of a lot of experimental or mechanism explanation data. It needs major revision before publication.

1.      First of all, the other adsorption methods mentioned in your article have disadvantages. Please show the advantages of your material compared to them.

2.      The structural characterization of the material is lacking. Only TEM data are seen, and the specific surface area and structure of the material are not clear yet. Please supplement other characterization data, such as SEM, BET, XRD, etc.

3.      In the material properties characterization section, you only do the adsorption efficiency of the material. The evaluation of the performance of adsorption materials should not only be in this aspect, but also supplement the maximum adsorption capacity, adsorption rate, adsorption kinetics of the material. The cycling performance of the material and the adsorption effect of other pollutants are also unknown. Please complete relevant experiments.

4.      You mentioned in your article that the adsorption effect of this material on neutral metal is different in a single system and a competitive system, and its mechanism is vague. Please discuss it in detail. The adsorbent has the best adsorption effect on heavy metal Cd, How about the selectivity of the material? In addition, the material you mentioned has better adsorption effect in the competitive system, which is also lacking in explanation. Please elaborate on the above issues.

5.      The magnification in TEM should be marked with a ruler, the performance graph in texture properties is a single color and the layout of the tables and pictures in the article is too messy and the size is not uniform. Please rearrange them and mark the serial numbers on the pictures or tables to make them intuitive and clear.

6.      There are some mistakes in grammar and format in this paper.

7.      The reference format is not unified in some places, so the format should be consistent. A related reference (Frontiers of Environmental Science & Engineering, 2020, 14 (4):68) on adsorption by biochar should be cited.

Reviewer 2 Report

The manuscript deals with the use of biochar materials to be used for the removal of heavy metals from aqueous solutions.

The approach of exploiting invasive vegetable species and waste organic materials as a starting material for the production of adsorbents for water decontamination is a topic attracting good attention. The performance obtained with some of the proposed materials seems to be interesting. However, in a manuscript for a chemical journal, it is essential to have a rigorous interpretation of the experimental results and a clear discussion on the data.

The present manuscript appears to be still in a very preliminary form. Several parts of the text need a careful revision, also in terms of the language (sorbets instead of sorbents; heterogonous instead of heterogeneous) or text formatting (Table 1 is really puzzling).

In deeper detail, many points do not reach the minimum quality standards of the journal:

1)      The nature as “nanomaterials” of the nano-biochars is not evident. Also the WT and WM materials, according to Fig. 1, show nanosized particles. So, according to which features WT and WH differ from NWT and NWH? Do the Authors have any further evidence about the “formation” of nanostructured materials? What is the “standard ceramic method” mentioned at line 120? Actually, Ref. 21 (cited there) refers to polycrystalline mixed oxides that are obtained by extensive grinding and thermal treatments, but mixed oxides are totally different from carbonaceous materials as the ones of biochars. So, a better definition of the method according to which biochars are transformed into biochars is needed.

2)      A better description of the methods and instruments used to determine EC and TDS is necessary. How were the total dissolved salts measured? By conductimetry? No mention of the use of a FT-IR instrument is reported either. The Experimental section on these points is too scarce.

3)      Section 3.1.3 is the weakest section of the entire text. The layout of the spectra is unusual. They are presented in transmittance, that typically goes from 0 to 100%, but in fig. 2 transmittance spans from 0 to 250. In addition, in figs. 2-5, it is not clear at all where peaks are located. Are the bands upwards or downwards? For instance, C=O band in Fig. 2 is placed around 1840 cm-1, where there is a maximum of transmittance (i.e. a minimum of absorbance), but this is a non-sense for an absorption band. Other “apparent” bands in fig 3 or 4 are not evident at all. So, the description in the main text does not follow the labels on the spectra. Moreover, some attributions are completely confusing: transition metal carbonyls? Where do M(CO)x come from, as they are typically unstable in water? How can C-F or C-Br or C=C=O functions be generated under these conditions? The entire section is confused and does not attain the minimum quality standards for the journal.

4)      In section 3.2., original data from Authors’ work and previous literature results are mixed and it is extremely difficult to understand the difference between the two. In addition, data expressed in terms of mg of heavy metal removed per gram of adsorbent could be useful to have an idea of the performance of the material.

5)      Figure 3 is poorly sketched. Four profiles are reported, but in the caption only two curves are described (WH and WT). What yellow and violet profile indicate? A better attention to the presentation of the experimental data is necessary.

Taking into account all these major problems, the contribution cannot be considered for publication. A complete re-writing of the text is necessary (in particular the FT-IR section) for a proper evaluation.